# CORE-3D: Context-aware Open-vocabulary Retrieval by Embeddings in 3D

## Abstract

3D scene understanding is fundamental for embodied AI and robotics, supporting reliable perception for interaction and navigation. Recent approaches achieve zero-shot, open-vocabulary 3D semantic mapping by assigning embedding vectors to 2D class-agnostic masks generated via vision-language models (VLMs) and projecting these into 3D. However, these methods often produce inaccurate semantic assignments due to the direct use of raw masks, limiting their effectiveness in complex environments. To address this, we leverage SemanticSAM with progressive granularity refinement to generate more accurate and numerous object-level masks. To further enhance semantic context, we employ a context-aware CLIP encoding strategy that integrates multiple contextual views of each mask, providing much richer visual context. We also propose a new VLM based framework to process complex 3D queries and retrieve target ojects based on 3D position relations between different objects. We evaluate our approach on multiple 3D scene understanding tasks, including 3D semantic segmentation and object retrieval from language queries, across several benchmark datasets. Experimental results demonstrate significant improvements over existing methods, highlighting the effectiveness of our approach.

## 1 Introduction

Accurate understanding of 3D environments at the object level is a fundamental requirement for embodied AI, robotics, and augmented/virtual reality applications (Anderson et al., 2018; Batra et al., 2020; Szot et al., 2021; Gu et al., 2024). Tasks such as robotic manipulation (Zeng et al., 2020; Xu et al., 2020) and autonomous navigation (Xu et al., 2020) depend on reliable 3D scene representations, while AR/VR systems require precise object-level maps to anchor virtual content in the physical world (Kerr et al., 2023). 3D semantic segmentation directly enables these capabilities by assigning category labels to each point in a scene, yielding dense and structured maps that support high-level reasoning and interaction (Qi et al., 2017). Beyond dense labeling, many practical applications require agents not only to segment and recognize objects but also to retrieve specific objects from natural language queries—for example, "find the chair closest to the table" or "locate the vase on the shelf"(Chen et al., 2020; Achlioptas et al., 2020). Such capabilities are essential for interactive agents operating in open and cluttered real-world environments.

Despite progress in supervised 3D scene understanding methods, constructing accurate 3D semantic maps in cluttered, real-world environments remains highly challenging, due to occlusions, incomplete observations, and the prohibitive cost of acquiring large-scale annotated 3D data(Patel et al., 2025; Yu et al., 2025). To reduce reliance on expensive 3D annotations, recent works (Gu et al., 2024; Jatavallabhula et al., 2023) have explored open-vocabulary 3D scene understanding by combining segmentation models with vision–language models. These approaches first extract object masks from 2D images using a segmentation backbone, then assign semantic embeddings to each mask by a vision–language model such as CLIP (Radford et al., 2021). Projecting these embedded masks into 3D yields semantic maps without requiring task-specific training which enables zero-shot labeling and responding to complex 3D language query tasks. Despite the benefits of these approaches, they encounter some challenges that should be accurately handled. First, 2D segmentation backbones such as SAM (Kirillov et al., 2023) often generate fragmented or incomplete masks, especially in cluttered indoor environments, leading to severe over-segmentation. Second, applying CLIP directly to individual masks provides limited semantic context. Third, aggregating predictions across multiple frames can introduce inconsistencies, as the same object may receive different

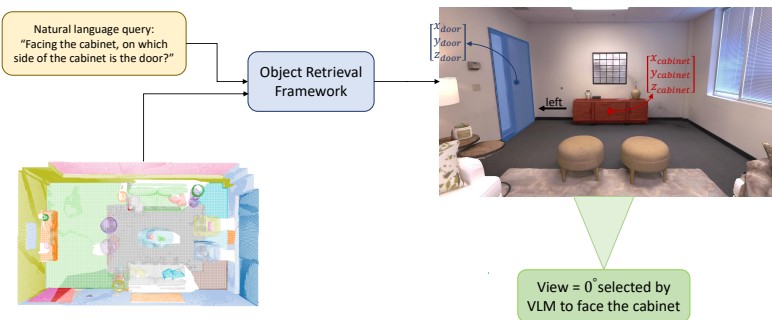

Figure 1: Illustration of object retrieval from natural language in a 3D scene. A natural language query specifies a target and spatial relation ("Facing the cabinet, on which side of the cabinet is the door?"). Our framework retrieves object embeddings, grounds them in 3D coordinates, selects the appropriate view to face the cabinet using VLM, and reasons about spatial orientation to output the correct relation.

contextual embeddings depending on viewpoint. As a result, existing foundation-model-based approaches still struggle to construct coherent and reliable 3D semantic maps.

In this work, we present a training-free pipeline that overcomes these challenges by improving both segmentation and embedding generation through progressive refinement and context-aware encoding. First, we leverage SemanticSAM (Li et al., 2023a) with progressive granularity adjustment to generate accurate and complete class-agnostic object-level masks, mitigating the fragmentation issues of vanilla SAM. Second, we introduce a context-aware CLIP encoding strategy that aggregates multiple complementary views of each mask with empirically chosen weighting, providing the semantic context necessary for robust classification. Finally, we enforce multi-view consistency by merging overlapping masks in 3D and filtering incomplete or spurious segments with geometric heuristics. Together, these components enable the construction of coherent, high-quality 3D semantic maps in an open-vocabulary, training-free setting, without requiring any 3D supervision.

Beyond 3D semantic segmentation, we extend our framework to object retrieval from natural language instructions. Queries are processed with a large language model (LLM) to extract the target and anchor categories and relational constraints (e.g., "nearest to the door" or "on top of the table"), which are then matched against our fused 3D object embeddings. We also leverage vision large language models to confirm the candidates of target and anchor objects and generate final output by processing the language query together with 3D positions of the confirmed objects. This enables retrieval grounded in both category semantics and spatial relations.

Our main contributions are:

- We introduce a SemanticSAM refinement strategy that incrementally adjusts granularity, yielding more accurate and complete masks than vanilla SAM.

- We propose a context-aware CLIP feature aggregation scheme that combines multiple contextual views of each mask to ensure robust open-vocabulary classification.

- We enforce reliable multi-view semantic 3D map by merging overlapping predictions in 3D using geometric and semantic heuristics.

- We perform the zero-shot 3D semantic segmentation on Replica (Straub et al., 2019) and ScanNet (Dai et al., 2017) datasets.

- We also extend the pipeline to open-vocabulary 3D object retrieval, using LLM-based query parsing to handle relational constraints in natural language, and evaluate this on the SR3D (Achlioptas et al., 2020) benchmark.

- Our method achieves superior results on Replica, ScanNet, and SR3D, outperforming prior open-vocabulary approaches in mIoU, f-mIoU, and mAcc for segmentation, while also demonstrating strong retrieval performance.

## 2 RELATED WORK

### 2.1 FOUNDATION MODELS FOR VISION–LANGUAGE ALIGNMENT

Large-scale vision–language models have become the cornerstone of open-vocabulary perception. Contrastive pretraining approaches such as CLIP (Radford et al., 2021) learn aligned image and text embeddings from massive web corpora, enabling zero-shot classification, retrieval, and multimodal reasoning without task-specific finetuning. Building on this paradigm, ALIGN (Jia et al., 2021) and Florence (Yuan et al., 2021) improved representation quality. Subsequently, OVSeg (Liang et al., 2023) addresses the CLIP bottleneck in two-stage segmentation by finetuning on masked regions and introducing mask prompt tuning. But even this pipeline can not fully resolve the problem of generating highly enriched embeddings for 2D masks.

### 2.2 FOUNDATION MODELS FOR MASK GENERATION

Complementing semantic embeddings, class-agnostic segmentation priors have shown remarkable generalization across diverse domains. MaskFormer (Cheng et al., 2021) unified semantic and instance segmentation by reformulating both tasks as per-pixel mask classification, while its successor Mask2Former (Cheng et al., 2022) introduced masked attention to achieve strong panoptic segmentation performance with improved efficiency. MaskDINO (Li et al., 2023b) further integrates detection and segmentation in a unified transformer, showing strong generalization to unseen categories. The Segment Anything Model (SAM) (Kirillov et al., 2023), trained on billions of masks, demonstrated that a single backbone can transfer across domains and serves as a universal prior for open-vocabulary pipelines. However, SAM often produces fragmented or incomplete masks in cluttered indoor scenes. SemanticSAM (Li et al., 2023a) alleviates this by generating Segmnet masks with different granularity levels from semantic to instance and to parts.

### 2.3 TRAINING-BASED 3D SCENE UNDERSTANDING

There are several ways to train a model for assigning embeddings to 3D points. LERF (Kerr et al., 2023) utilizes the structure of NeRF (Mildenhall et al., 2020) models to assign not only colors and 3D positions, but also embedding vectors generated by the CLIP model to 3D points. While Open-NeRF (Engelmann et al., 2024) leverages OpenSeg (Ghiasi et al., 2022) feature maps to supervise the 3D feature maps.
Some models blend this idea with 3D Gaussian Splatting (Kerbl et al., 2023) structure. LangSplat (Qin et al., 2024) uses CLIP embeddings of SAM-generated masks to train an encoder–decoder architecture that produces low-dimensional semantic features for 3D Gaussian distributions. Open-Gaussian (Wu et al., 2024), on the other hand, introduces an intra-mask smoothing loss and an inter-mask contrastive loss to enhance performance on 3D semantic segmentation tasks.

### 2.4 ZERO-SHOT 3D SCENE UNDERSTANDING

Foundation models provide strong 2D priors, and recent works extend open-vocabulary perception into 3D settings, which is crucial for robotics and embodied AI. A key trend is integrating vision–language models with 3D representations for mapping, scene understanding, and grounding. ConceptFusion (Jatavallabhula et al., 2023) introduces open-set 3D mapping by fusing image features with 3D reconstructions for dense semantic labeling of novel concepts. ConceptGraphs (Gu et al., 2024) propose open-vocabulary 3D scene graphs that align CLIP features with geometry, supporting perception and planning. VoxPoser (Huang et al., 2023) applies LLMs and VLMs to synthesize 3D value maps, enabling zero-shot, open-set robot manipulation. The Open-Vocabulary Octree-Graph (Wang et al., 2024) uses adaptive octrees to encode occupancy and semantics compactly, while Beyond Bare Queries (BBQ) (Linok et al., 2025) leverages 3D scene graphs and LLM reasoning for precise language-conditioned object retrieval. For robotics, Hierarchical Open-Vocabulary 3D Scene Graphs (HOV-SG) (Werby et al., 2024) construct hierarchical floor-room-object graphs

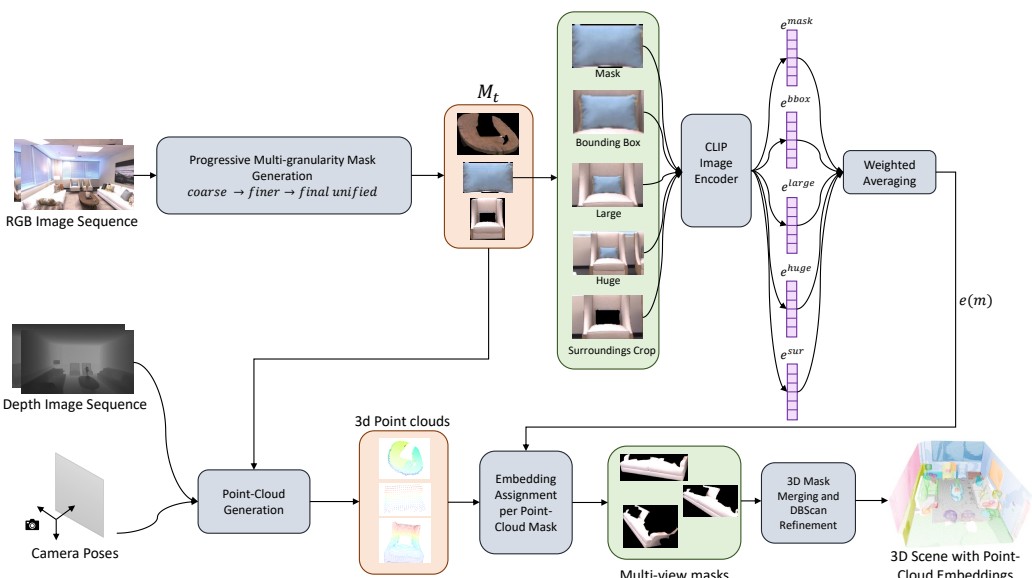

Figure 2: Overview of our training-free open-vocabulary 3D semantic segmentation and retrieval pipeline. Given RGB–D image sequences, we first generate progressive multi-granularity 2D masks ($M_t$) to mitigate fragmentation. Each mask is encoded with CLIP using multiple contextual crops (mask, bounding box, large, huge, surroundings), and their embeddings are aggregated via weighted averaging. In parallel, depth maps and poses are fused into a 3D point cloud, where embeddings are assigned per point-cloud mask. Multi-view predictions are merged and refined with DBSCAN clustering to enforce consistency, resulting in a coherent 3D semantic map with point-cloud embeddings that support both open-vocabulary segmentation and object retrieval.

for long-horizon language-grounded navigation. In 2D, Pixels-to-Graphs (PGSG) (Li et al., 2024) generates scene graphs from images using a generative VLM, supporting both novel relations and downstream vision–language tasks. These works collectively highlight the importance of combining open-vocabulary semantics with 3D or 2D representations to advance perception, mapping, manipulation, and robot interaction.

## 3    METHOD

Let $\{(I_t, D_t, \mathbf{T}_t)\}_{t=1}^{T}$ denote a sequence of RGB images $I_t \in \mathbb{R}^{H \times W \times 3}$, depth maps $D_t \in \mathbb{R}^{H \times W}$, and corresponding camera poses $\mathbf{T}_t \in SE(3)$, where each pose encodes the position and orientation (roll, pitch, yaw) of the camera at time step $t$. Given $D_t$ and $\mathbf{T}_t$, each pixel $p = (u, v)$ can be back-projected into a unique 3D point $\mathbf{x}_p \in \mathbb{R}^3$, thereby allowing us to reconstruct the 3D environment across frames (Curless & Levoy, 1996; Hartley & Zisserman, 2003; Newcombe et al., 2011). Our framework leverages this setup to progressively build semantically meaningful 3D object representations from raw multi-view observations. Starting from 2D instance masks obtained with a refined segmentation strategy, we compute context-aware embeddings that integrate both object-level details and surrounding visual cues. These per-view masks and embeddings are then lifted into 3D, where multi-view consistency and spatial clustering are enforced to merge redundant detections and split over-merged instances. The resulting 3D object candidates are associated with unified embeddings that enable open-vocabulary semantic labeling via CLIP text-image alignment. Finally, we support natural-language object retrieval by grounding query semantics in the labeled 3D scene, allowing objects to be localized based on category, context, and orientation cues.

### 3.1    MASK GENERATION

Instead of relying on vanilla SAM (Kirillov et al., 2023) for 2D instance segmentation, which often produces *fragmented masks* in cluttered indoor scenes, we employ SemanticSAM (Li et al., 2023a), a

variant of SAM that exposes a *granularity parameter* $g \in \mathbb{R}^+$ controlling the scale of segmentation. Small values of $g$ yield coarse masks, while larger values produce finer-grained segments. However, using a single granularity level is suboptimal: coarse values may miss small objects, whereas fine values tend to over-segment large objects into multiple inconsistent parts.

To address this, we introduce a progressive refinement strategy. For each image $I_t$, we generate segmentations at an increasing sequence of granularity levels $\{g_1, g_2, \ldots, g_K\}$. At each step $k$, SemanticSAM produces a set of candidate masks, but we only keep the set of masks $N_k$ with more than a specific threshold of certainty $\tau_{cer}$ denoted as

$$\mathcal{M}_t^{(k)} = \{m_{t,1}^{(k)}, \ldots, m_{t,N_k}^{(k)}\},$$

where $N_k$ is the total number of masks generated at granularity level $g_k$. We then retain only those masks whose area has less than a threshold overlap with any mask discovered at previous levels:

$$\hat{\mathcal{M}}_t^{(k)} = \Big\{ m \in \mathcal{M}_t^{(k)} \mid \max_{m' \in \cup_{j<k} \hat{\mathcal{M}}_t^{(j)}} \frac{|m \cap m'|}{|m|} < \tau_k \Big\},$$

where $\tau_k \in [0,1]$ denotes the overlap threshold used when adding masks from granularity level $g_k$. In practice, $\tau_k$ is varied across levels to balance redundancy removal and coverage: stricter thresholds are applied at coarser levels, while more permissive thresholds are used at finer levels.

This procedure ensures that each new granularity level contributes *novel object candidates* without introducing redundant fragments. Intuitively, large objects are captured at coarse levels, while fine details and small objects are progressively added at higher granularity. By enforcing the threshold $\tau$ and applying these additional filters, we prevent duplicated or noisy masks, leading to a more accurate and complete set of object proposals.

When a single 2D mask spans multiple distinct objects that are spatially close in the 2D image but separated in 3D space (e.g., a vase in front of a couch), we refine it by applying DBSCAN clustering (Ester et al., 1996) to each $\hat{\mathcal{M}}_t^{(k)}$ to get the set of $\tilde{\mathcal{M}}_t^{(k)}$. This step separates the projected 3D points of $m$ into distinct clusters, each corresponding to a potential object. Each cluster is treated as a new candidate instance: we project it back to the image plane, generate the corresponding 2D mask. The final mask set for frame $t$ is then

$$\mathcal{M}_t = \bigcup_{k=1}^{K} \tilde{\mathcal{M}}_t^{(k)}.$$

## 3.2 CONTEXT-AWARE CLIP EMBEDDING

Given the refined 2D masks $\mathcal{M}_t$, we next compute semantic embeddings for each object candidate. A direct approach would be to crop the mask region and feed it into CLIP (Radford et al., 2021) or passing a raw 2D mask to OvSeg which is fine tuned on raw masks datasets. However, CLIP relies heavily on visual context, and isolated object crops often lead to ambiguous or incorrect embeddings. However, OvSeg struggles to overcome this issue, but in some scenarios, even an accurate mask does not contain meaningful semantics as shown in Fig. 2. To mitigate this, we construct a set of complementary visual crops for each mask that balance object detail with surrounding scene context.

Specifically, for each mask $m$, we extract five complementary crops from the RGB frame $I_t$: (i) **mask crop** ($I^{\text{mask}}$), where pixels outside the mask are set to zero; (ii) **bounding box crop** ($I^{\text{bbox}}$), the tight bounding box enclosing the mask; (iii) **large-context crop** ($I^{\text{large}}$), an expanded bounding box with scale factor 2.5; (iv) **huge-context crop** ($I^{\text{huge}}$), an expanded bounding box with scale factor 4; and (v) **surroundings crop** ($I^{\text{sur}}$), obtained by expanding the bounding box with scale factor 3 and blacking out the mask itself so that only the surrounding environment is visible.

Each of these crops is passed through the CLIP image encoder to obtain embeddings:

$$\mathbf{e}^{\text{mask}}, \; \mathbf{e}^{\text{bbox}}, \; \mathbf{e}^{\text{large}}, \; \mathbf{e}^{\text{huge}}, \; \mathbf{e}^{\text{sur}} \in \mathbb{R}^d,$$

where $d$ is the CLIP embedding dimension.

We then compute a context-aware embedding for mask $m$ by taking a weighted combination of these representations:

$$\mathbf{e}(m) = w_{\text{mask}} \mathbf{e}^{\text{mask}} + w_{\text{bbox}} \mathbf{e}^{\text{bbox}} + w_{\text{large}} \mathbf{e}^{\text{large}} + w_{\text{huge}} \mathbf{e}^{\text{huge}} - w_{\text{sur}} \mathbf{e}^{\text{sur}},$$

where the weights $\{w_{\text{mask}}, w_{\text{bbox}}, w_{\text{large}}, w_{\text{huge}}, w_{\text{sur}}\}$ are empirically tuned. Note that the surroundings embedding is subtracted with negative weight, enforcing contrastive context by penalizing features dominated by the environment rather than the object itself.

Finally, the embedding is normalized:

$$\mathbf{e}(m) \leftarrow \frac{\mathbf{e}(m)}{\|\mathbf{e}(m)\|_2}.$$

These per-view embeddings serve as initial semantic descriptors. During the subsequent 3D merging step, embeddings corresponding to the same physical object observed across multiple views are averaged to form unified object-level representations.

### 3.3 3D Mask Merging and Refinement

Projecting all pixels of mask $m$ yields a 3D point set $\mathcal{X}(m)$, from which we compute its volumetric occupancy $V(m)$ using a voxelization procedure. To consolidate multi-view observations, we evaluate the volumetric intersection between two candidate masks $m_a$ and $m_b$ as

$$\text{IoV}(m_a, m_b) = \frac{\text{Vol}(\mathcal{X}(m_a) \cap \mathcal{X}(m_b))}{\text{Vol}(\mathcal{X}(m_a))}, \qquad \text{IoV}(m_b, m_a) = \frac{\text{Vol}(\mathcal{X}(m_a) \cap \mathcal{X}(m_b))}{\text{Vol}(\mathcal{X}(m_b))}.$$

We merge $m_a$ and $m_b$ into a single 3D object if and only if the following conditions are satisfied:

$$\text{IoV}(m_a, m_b) > \gamma, \quad \text{IoV}(m_b, m_a) > \gamma, \quad \text{and} \quad |\text{IoV}(m_a, m_b) - \text{IoV}(m_b, m_a)| < \delta,$$

where $\gamma \in [0, 1]$ is the minimum overlap threshold and $\delta \in [0, 1]$ limits the allowable asymmetry between the two ratios.

This *symmetric–balanced IoV criterion* ensures that two masks are merged only when they exhibit both high mutual overlap and comparable volumetric support. It prevents degenerate cases where one object is almost fully contained within another but not vice versa—for example, a small cushion lying on a large couch—by rejecting merges with large asymmetry in overlap.

Along with merging their point clouds, we also average their embeddings:

$$\mathbf{e}_{\text{merged}} = \frac{1}{n} \sum_{i=1}^{n} \mathbf{e}(m_i),$$

where $m_1, \ldots, m_n$ denote masks that have been merged.

and re-apply our multi-crop CLIP embedding procedure. This ensures that each physically distinct object obtains its own semantic descriptor, even if they were originally fused into a single 2D segmentation.

After applying merging, we obtain the final set of refined 3D masks:

$$\mathcal{M}^{3D} = \{M_1^{3D}, M_2^{3D}, \ldots, M_N^{3D}\},$$

where each $M_i^{3D}$ is associated with a unified point cloud and an averaged embedding $\mathbf{e}(M_i^{3D})$.

### 3.4 Object Retrieval

We extend our pipeline to natural-language object retrieval, where the goal is to localize the specific instance referenced by a free-form query $q$. The task requires reasoning over objects, their spatial relations, and their orientation cues. Our retrieval pipeline consists of four stages.

**Query structuring.** We first convert the input query $q$ into a structured form

$$\Pi(q) = (m, \; \mathcal{R}, \; \Omega),$$

where $m$ denotes the name and attributes of the main object , $\mathcal{R}$ is a set of referenced object names and $\Omega$ encodes orientation constraints (e.g., "front of the cabinet").A lightweight LLM extractor produces $\Pi(q)$ deterministically; for the full prompt, see C.1.

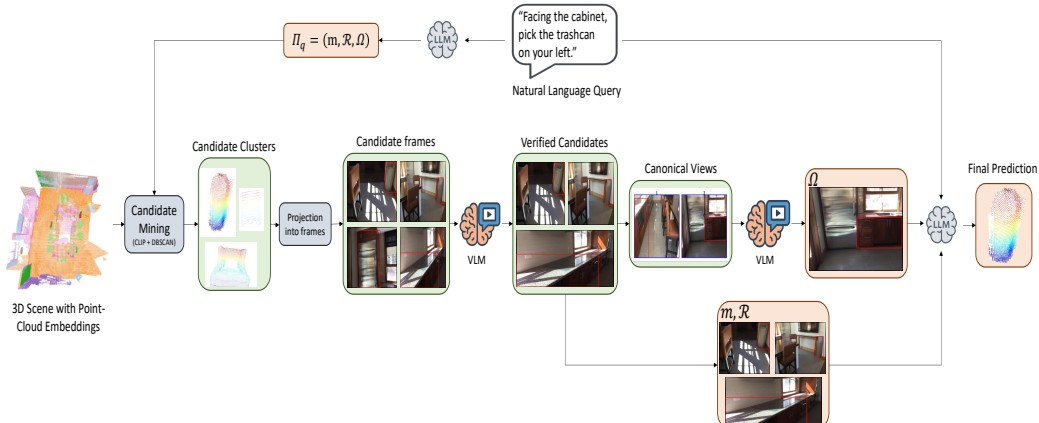

Figure 3: Pipeline for natural-language object retrieval. A free-form query is parsed into structured form $\Pi(q) = (m, \mathcal{R}, \Omega)$. Candidate objects are mined using CLIP similarity and DBSCAN clustering, projected into frames, and verified by a VLM restricted to bounding boxes. If orientation constraints $\Omega$ are present, canonical views are rendered and resolved with a VLM. Finally, an LLM reasons over the verified candidates, referenced objects $\mathcal{R}$, and orientation cues to select the final prediction.

**Candidate mining.** For each object name $x \in \{m\} \cup \mathcal{R}$, we compute CLIP similarity using pre-computed object embeddings and retain the top-$K$ matches, where $K$ is either a fixed hyperparameter or an upper bound predicted by an LLM based on the object category and scene context. Each match corresponds to a 3D point cluster. We deliberately avoid thresholding CLIP scores and instead always select the top-$K$, since a single similarity threshold that works well for one query can be unreliable across scenes with substantially different similarity distributions. To remove duplicates, we voxelize all candidate clusters and discard any cluster whose voxelized support overlaps a larger neighbor beyond a fixed ratio.

**View selection and VLM verification.** Each candidate cluster is then projected into the set of RGB frames. We select the frame that maximizes 3D–2D overlap while penalizing occlusion by other candidate clusters. A VLM is prompted with the bounding box region and asked a binary question to determine whether the object is present. Only candidates passing this check are retained; for the prompt used in this verification step, see C.2.

**Orientation grounding.** If $\Omega$ specifies an orientation, we collect canonical views of each candidate at discretized yaw bins. These views are tiled into a numbered grid, and a VLM is asked to select the index corresponding to the orientation token (e.g., `front`). The chosen index is mapped back to a yaw angle and stored with the candidate; for the prompt used in this step, see C.3.

**Final reasoning.** The remaining candidates for $m$, together with centroids of related objects and any orientation cues, are passed to an LLM. The LLM receives the original query and structured scene geometry and outputs the index of the final prediction; for the prompt used in this step, see C.4.

## 4 EXPERIMENTS

We evaluate our framework on two tasks: (i) 3D open-vocabulary semantic segmentation, where the goal is to assign category labels to 3D object instances without task-specific training, and (ii) natural-language object retrieval, where the goal is to localize objects in a 3D scene given free-form text queries that may contain relational and orientation constraints. All experiments are conducted on a workstation equipped with a single NVIDIA RTX4090, with the exception of the VLM and LLM components, which are accessed through external APIs.

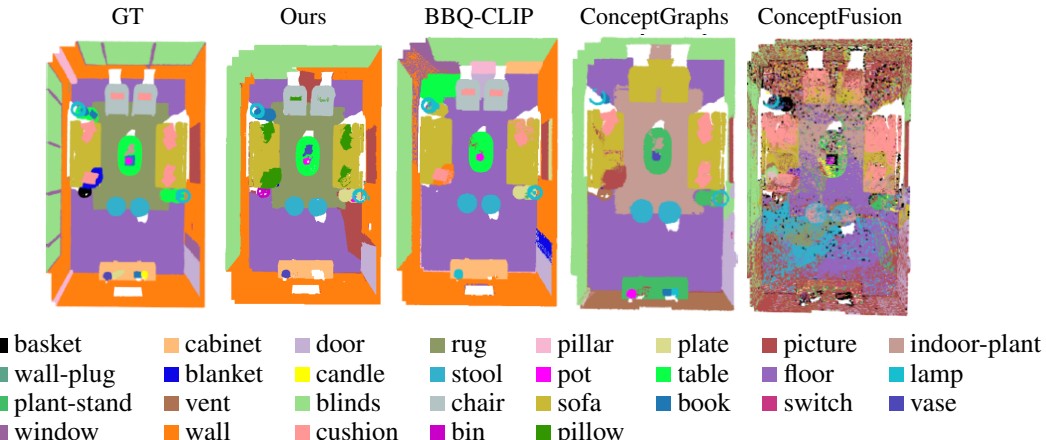

| GT | Ours | BBQ-CLIP | ConceptGraphs | ConceptFusion |

- ■ basket
- ■ cabinet
- ■ door
- ■ rug
- ■ pillar
- ■ plate
- ■ picture
- ■ indoor-plant
- ■ wall-plug
- ■ blanket
- ■ candle
- ■ stool
- ■ pot
- ■ table
- ■ floor
- ■ lamp
- ■ plant-stand
- ■ vent
- ■ blinds
- ■ chair
- ■ sofa
- ■ book
- ■ switch
- ■ vase
- ■ window
- ■ wall
- ■ cushion
- ■ bin
- ■ pillow

Figure 4: Qualitative comparison of 3D open-vocabulary semantic segmentation on Replica scenes. The GT, BBQ-CLIP, ConceptGraphs, OpenFusion, ConceptFusion columns are adapted from Linok et al. (2025), reproduced here under fair use for research comparison. Our method yields more accurate segmentation boundaries and finer recognition of challenging categories; notably, it is the only method that segments and labels the rug correctly, a class frequently missed or confused by competing approaches.

## 4.1 3D OPEN-VOCABULARY SEMANTIC SEGMENTATION

**Datasets.** We conduct experiments on two standard benchmarks: Replica (Straub et al., 2019) and ScanNet (Dai et al., 2017). Replica provides high-quality synthetic RGB-D scans of indoor environments with accurate ground-truth meshes and semantic annotations, while ScanNet consists of large-scale real-world RGB-D sequences with manually annotated 3D semantic and instance labels. Following prior work, we use eight Replica scenes: `room0`, `room1`, `room2`, `office0`, `office1`, `office2`, `office3`, and `office4`, and eight ScanNet scenes: `0011_00`, `0030_00`, `0046_00`, `0086_00`, `0222_00`, `0378_00`, `0389_00`, and `0435_00`. This subset selection ensures comparability with previous zero-shot methods. For text–image alignment, we use the Eva02-L CLIP (Fang et al., 2023; Yang et al., 2023) vision-language model.

**Baselines.** We compare against both training-based and zero-shot 3D open-vocabulary segmentation methods. Training-based baselines include LERF (Kerr et al., 2023), OpenNeRF (Engelmann et al., 2024), LangSplat (Qin et al., 2024), and OpenGaussian (Wu et al., 2024). Zero-shot baselines include ConceptFusion (Jatavallabhula et al., 2023), ConceptGraphs (Gu et al., 2024), BBQ-CLIP (Linok et al., 2025), OpenMask3D (Takmaz et al., 2023), and HOV-SG (Werby et al., 2024). Note that HOV-SG reports results on a different subset of ScanNet, so we omit its numbers for fair comparison.

**Results.** Table 1 reports quantitative comparisons on Replica and ScanNet. Our method achieves the best performance across all evaluated approaches—including both training-based and zero-shot methods. In particular, the improvements in mIoU and fwIoU highlight the benefit of our context-aware embeddings and multi-view 3D refinement, which produce more consistent object representations than 2D-based pipelines. We also provide qualitative examples in Fig. 4. Visual comparisons on Replica scenes illustrate that our method yields more accurate segmentation boundaries and finer recognition of challenging categories. Our method consistently detects objects across categories with higher fidelity than competing approaches.

## 4.2 NATURAL-LANGUAGE OBJECT RETRIEVAL

**Datasets.** We evaluate on the Sr3D+ benchmark (Achlioptas et al., 2020), which provides diverse referring expressions such as relational and orientation-based queries (e.g., "the table that is far from the armchair" or "Facing the cabinet, pick the trashcan on your left."). Following the BBQ (Linok et al., 2025) setup, we use the same 661 sampled instructions. Each query is paired with a

Table 1: 3D open-vocabulary semantic segmentation benchmark.

| Methods | Replica | | | ScanNet | | |
|---|---|---|---|---|---|---|
| | mAcc↑ | mIoU↑ | fmIoU↑ | mAcc↑ | mIoU↑ | fmIoU↑ |
| *Training-based methods* | | | | | | |
| LERF | 0.26 | 0.11 | – | – | – | – |
| OpenNeRF | 0.32 | 0.20 | – | – | – | – |
| LangSplat | – | – | – | 0.09 | 0.04 | – |
| OpenGaussian | – | – | – | 0.42 | 0.25 | – |
| *Zero-shot methods* | | | | | | |
| ConceptFusion | 0.29 | 0.11 | 0.14 | 0.49 | 0.26 | 0.31 |
| OpenMask3D | – | – | – | 0.34 | 0.18 | 0.20 |
| ConceptGraphs | 0.36 | 0.18 | 0.15 | 0.52 | 0.26 | 0.29 |
| HOV-SG | 0.30 | 0.23 | 0.39 | – | – | – |
| BBQ-CLIP | **0.38** | 0.27 | 0.48 | 0.56 | 0.34 | 0.36 |
| Ours | **0.38** | **0.29** | **0.56** | **0.61** | **0.36** | **0.46** |

Table 2: **Grounding accuracy on Sr3D+.** Accuracy at IoU thresholds A@0.1 and A@0.25 across subsets: Easy, Hard, View-dependent, and View-independent.

| Methods | Overall | | Easy | | Hard | | View Dep. | | View Indep. | |
|---|---|---|---|---|---|---|---|---|---|---|
| | A@0.1 | A@0.25 | A@0.1 | A@0.25 | A@0.1 | A@0.25 | A@0.1 | A@0.25 | A@0.1 | A@0.25 |
| OpenFusion | 12.6 | 2.4 | 14.0 | 2.4 | 1.3 | 1.3 | 3.8 | 2.5 | 13.7 | 2.4 |
| BBQ-CLIP | 14.4 | 8.8 | 15.4 | 9.0 | 6.7 | 6.7 | 11.4 | 5.1 | 14.4 | 8.8 |
| ConceptGraphs | 13.3 | 6.2 | 13.0 | 6.8 | 16.0 | 1.3 | 15.2 | 5.1 | 13.1 | 6.4 |
| BBQ | 34.2 | 22.7 | 34.3 | 22.7 | 33.3 | 22.7 | **32.9** | 20.3 | 34.4 | 23.0 |
| **Ours** | **41.8** | **35.6** | **41.8** | **35.7** | **41.3** | **34.7** | **32.9** | **30.4** | **43.0** | **36.3** |

ground-truth target (GT) and labeled as *Easy*, *Hard*, *View-dependent*, or *View-independent*, allowing systematic evaluation across reasoning challenges.

**Baselines.** We compare against recent zero-shot 3D grounding approaches, including OpenFusion (Yamazaki et al., 2024), BBQ-CLIP (Linok et al., 2025), ConceptGraphs (Gu et al., 2024), and BBQ (Linok et al., 2025).

**Results.** Table 2 summarizes quantitative results. Our method substantially outperforms all baselines across both IoU thresholds and all difficulty subsets.

## 5 ABLATION STUDIES

In this section, we assess the effectiveness of each part of the 3D semantic segmentation pipeline and also of 3D object retrieval. First of all, we have evaluated our progressive multi granularity level mask generation strategy. We compare our approach with Vanilla SAM model and also with different single granularity levels of Semantic-SAM model.

Table 3: Granularity-based mask generation impact

| Methods | Replica | | | ScanNet | | |
|---|---|---|---|---|---|---|
| | mAcc↑ | mIoU↑ | fmIoU↑ | mAcc↑ | mIoU↑ | fmIoU↑ |
| SAM | 0.32 | 0.22 | 0.37 | 0.43 | 0.27 | 0.42 |
| Semantic-SAM(Granularity Level=2) | 0.34 | 0.25 | 0.51 | 0.54 | 0.32 | 0.43 |
| Semantic-SAM(Granularity Level=4) | **0.38** | 0.24 | 0.43 | 0.43 | 0.26 | **0.46** |
| Semantic-SAM(Granularity Level=6) | 0.26 | 0.11 | 0.14 | 0.17 | 0.09 | 0.16 |
| Ours | **0.38** | **0.29** | **0.56** | **0.61** | **0.36** | **0.46** |

To evaluate the effectiveness of our context-aware CLIP embedding generation approach, compare it to passing the raw masks to the OvSeg (Liang et al., 2023) model which is the finetuned version of the CLIP model on the raw masks dataset.

Table 4: Context-aware CLIP embedding impact

| Methods | Replica | | | ScanNet | | |
|---|---|---|---|---|---|---|
| | mAcc↑ | mIoU↑ | fmIoU↑ | mAcc↑ | mIoU↑ | fmIoU↑ |
| OvSeg | 0.21 | 0.11 | 0.42 | 0.27 | 0.16 | 0.21 |
| Context-Aware CLIP | **0.38** | **0.29** | **0.56** | **0.61** | **0.36** | **0.46** |

For our approach, we have used Large version of CLIP model which has the same number of parameters as the finetuned OvSeg model. Our approach strongly enhances the CLIP model's embeddings quality without any finetuning.

We have also evaluated our extension mechanism. Instead of relying on target mask size for extending the view around the target mask, we can leverage neighbor masks sizes to determine extension ratio. We define the large extension ration as the ratio that covers all the neighboring masks and the Huge as the ratio that covers all the neighbors of neighboring masks.

Table 5: Extension mechanism impact.

| Methods | Replica | | | ScanNet | | |
|---|---|---|---|---|---|---|
| | mAcc↑ | mIoU↑ | fmIoU↑ | mAcc↑ | mIoU↑ | fmIoU↑ |
| Neighbor coverage | 0.34 | 0.28 | 0.45 | 0.51 | 0.28 | 0.36 |
| Mask size | **0.38** | **0.29** | **0.56** | **0.61** | **0.36** | **0.46** |

## 6 CONCLUSION

We introduced CORE-3D, a training-free pipeline for open-vocabulary 3D perception that combines progressive SemanticSAM refinement, context-aware CLIP embeddings, and multi-view 3D consolidation. This design reduces mask fragmentation, preserves semantic context, and yields coherent object-level maps without requiring 3D supervision. Experiments on Replica and ScanNet show consistent gains in mIoU and fmIoU, while on Sr3D+ our retrieval pipeline—based on structured parsing, VLM verification, and geometric reasoning—achieves clear improvements in grounding accuracy. Our results suggest that leveraging richer 2D segmentation and embedding strategies is a powerful alternative to supervision-heavy pipelines, especially in cluttered, open-world environments. Beyond segmentation and retrieval, extending the framework with temporal consistency and deeper integration with multimodal reasoning models could further enhance robustness and generality. In sum, CORE-3D demonstrates that careful refinement and context-rich embeddings make zero-shot 3D mapping and language-grounded retrieval both practical and reliable.

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

# A  3D SEMANTIC SEGMENTATION

## A.1  LABELING PROTOCOL

For open-vocabulary labeling, we use the standard CLIP paradigm (Radford et al., 2021): each refined 3D object mask is encoded into an embedding and compared against a set of text embeddings corresponding to candidate categories. The category with the highest similarity is assigned as the predicted label. This procedure enables zero-shot semantic segmentation without task-specific training, while our multi-view refinement makes the assignments more robust to occlusions and clutter.

## A.2  EVALUATION PROTOCOL

We follow the evaluation protocol in prior work. For each predicted point cloud instance, we assign a semantic label by finding the nearest ground-truth instance (based on centroid distance) and transferring its label. For each scene, we restrict the text prompts to the classes present in the ground-truth annotations, formatted as "a photo of `<class name>`." We report mean accuracy (mAcc), mean intersection-over-union (mIoU), and frequency-weighted mIoU (fmIoU).

# B  3D OBJECT RETRIEVAL

## B.1  IMPLEMENTATION DETAILS

We use the EVA02 CLIP backbone (Fang et al., 2023) for visual–text alignment, and `qwen2.5-vl-32b-instruct` as the vision–language model for multimodal encoding. Our first LLM, `gpt-5-mini`, is employed for object extraction, while the second LLM, `openai-o4-mini`, handles the final decision-making stage.

## B.2  EVALUATION PROTOCOL

Following prior work, we report grounding accuracy at two IoU thresholds: A@0.1 and A@0.25. A prediction is considered correct if the IoU between the predicted and ground-truth bounding box exceeds the threshold. Accuracy is reported overall as well as separately for the four difficulty subsets.

# C  LLM PROMPTS FOR OBJECT RETRIEVAL

In this work, we use both LLMs and VLMs to interpret natural-language queries and convert them into structured outputs for 3D object retrieval. In this section, we provide the prompts used in our system, covering query parsing, main-object extraction, and candidate selection.

## C.1  QUERY PARSING

The retrieval pipeline begins by converting each natural-language instruction into a structured representation containing the main object, related objects, the spatial relation between them, and any orientation constraints. This is obtained using the following prompt:

> Your task is to strictly extract structured information from the given text.
> Format rules (follow exactly): 1) Line 1 — MAIN OBJECT: write only one object (the target), optionally followed by comma-separated visual attributes (color, shape, size, texture, material). No non-visual or spatial attributes.
> 2) Line 2 — RELATED OBJECTS: list all related objects with their visual attributes using: object_name[, attr1, attr2...]; object_name2[, attr1...] Do not repeat the main object. No spatial attributes. Leave blank if none.
> 3) Line 3 — RELATION KEYWORD: write exactly one of: closest, farthest, left, right, front, back, below, supported-by, above, supporting, between
> 4) Line 4 — ORIENTATION IMPORTANCE: list objects with required orientation in the form: object_name: orientation Allowed: front, back, left, right. Leave blank if none.
> Do not add placeholders, extra text, labels, punctuation, or formatting.
> Text: `<user_instruction>`
> Return exactly four lines.

This prompt provides a deterministic mapping from free-form language to a structured form suitable for geometric reasoning.

Although the prompt requests a spatial-relation keyword on Line 3, this information is not used by our retrieval pipeline. It is included only for analysis, and can be removed without affecting the retrieval system itself.

## C.2 OBJECT PRESENCE DETERMINATION

After selecting the appropriate frame for each candidate, a red bounding box is drawn around its projected 2D region, and the annotated frame is provided to the VLM together with the following prompt:

> Focus only on the red bounding box in the image. Ignore everything outside the box. Determine if the object `<object_name>` is present inside the box. It is valid even if only part of the object appears, as long as the main subject is included. Answer strictly with 'Yes' or 'No'.

This prompt ensures that the VLM evaluates the candidate purely based on the content of the region indicated by the bounding box, while visually ignoring the rest of the frame.

## C.3 VIEWPOINT IDENTIFICATION

For instructions that include orientation-dependent relations (such as "front of," "left of," or "to the right when facing the cabinet"), the system must determine which viewpoint of the candidate object corresponds to the requested orientation. To do this, we render viewpoints at sufficiently separated yaw angles (e.g., 90° increments), determined by camera locations relative to the object. These views are arranged into a single grid image, with each sub-image labeled by an index and containing a red bounding box marking the target object.

This grid image is then given to the VLM, which selects the sub-image matching the orientation specified in the instruction. The prompt used is:

> In the provided image, multiple sub-images are arranged in a grid, each bordered in blue and labeled with a number at the top. In every sub-image, the target object '`<object_name>`' is indicated with a red bounding box. Ignore all other content. Determine which sub-image shows the object from the '`<orientation>`' view. Output only the index number above that sub-image.

This allows the VLM to choose the correct viewpoint among all candidates presented within the same image grid.

## C.4 CANDIDATE SELECTION AND FINAL RETRIEVAL

Once presence verification and viewpoint identification (when required) are completed, the system must choose the correct 3D instance of the main object. To do so, an LLM is provided with the original instruction together with the 3D centroids of all candidate object clusters. Each centroid corresponds to the geometric center of the candidate's projected 3D point cloud. The LLM receives these centroids in plain text form along with the positions of referenced objects and, when applicable, the estimated orientation angles.

Depending on whether the instruction requires orientation reasoning, one of the two prompts below is used.

**Without orientation.** If the instruction does not involve orientation-dependent relations, the LLM receives the following prompt:

> Text: `<user_instruction>`
> The main object is `<main>`. It has the following candidate positions with indices (x, y, z):
> Index 0: (x=`<x0>`, y=`<y0>`, z=`<z0>`)
> Index 1: (x=`<x1>`, y=`<y1>`, z=`<z1>`)
> Index 2: (x=`<x2>`, y=`<y2>`, z=`<z2>`)
> . . .
> Other objects and their positions are:
> `<obj>`[`<k>`]: (x=`<x>`, y=`<y>`, z=`<z>`)
> . . .
> Instruction: Based on the text above, decide which candidate index of `<main>` is the correct target. Return only the numeric index (0, 1, 2, . . . ). Do not return anything else.

Here, all coordinate placeholders (e.g., `<x0>`, `<y>`) are filled with the centroids of the 3D clusters, and name placeholders (e.g., `<main>`, `<obj>`) are filled with the corresponding object identifiers.

**With orientation.** If the instruction requires orientation-dependent reasoning, the LLM receives the same information plus explicit orientation angles and their derived directional axes. The exact prompt is:

> Text: `<user_instruction>`
> The main object is `<main>`.
> It has the following candidate positions with indices (x, y, z where z is height):
> Index 0: (x=`<x0>`, y=`<y0>`, z=`<z0>`)
> Index 1: (x=`<x1>`, y=`<y1>`, z=`<z1>`)
> . . .
> Other objects and their positions are:
> `<obj>`[`<k>`]: (x=`<x>`, y=`<y>`, z=`<z>`)
> . . .
>
> Orientation information (object, label) and angles in degrees:
> `<obj>` (`<lab>`): [`<ang>`] When facing `<lab>` (`<ang>`°): Forward = `<ang>`° (`<dirF>`), Left = `<left>`° (`<dirL>`), Right = `<right>`° (`<dirR>`), Back = `<back>`° (`<dirB>`).
> Instruction: Based on the text above, decide which candidate index of `<main>` is the correct target. Return only the numeric index (0, 1, 2, . . . ). Do not return anything else.

Here, coordinate placeholders (e.g., `<x0>`, `<x>`, `<y>`, `<z>`) are filled with 3D centroids, name placeholders (e.g., `<main>`, `<obj>`) are filled with object identifiers, and orientation placeholders (e.g., `<lab>`, `<ang>`, `<dirF>`, `<left>`, `<right>`, `<back>`) are filled with the yaw angle and its derived directional axes.

In both cases, the LLM selects the final target purely from this structured geometric information and outputs a single integer index.

# D  MASK MERGING

We use weighted averaging for our mask merging part. The weight of each mask is determined by its 3D relative position from the camera, amount of semantic the mask contains and the number of pixels in the mask. The position weight is determined by this formula:

$$w_d = \begin{cases} e^{-\alpha(1.5-d_{avg})} & \text{if } d_{avg} \leq 1.5, \\ 1.0 & \text{if } 2.5 \geq d_{avg} \geq 1.5, \\ e^{-\alpha(d_{avg}-2.5)} & \text{if } d_{avg} < 2.5 \end{cases}$$

The $d_{avg}$ is the average depth of pixels in the mask and $\alpha = 0.5$.
To measure semantics of each mask, we measure the cosine similarity of each mask with a set of indoor classes texts. The set contains 200 indoor classes and is generated by an LLM. The semantic weight is defined as $w_s = \max_{s \in S} s$.

The $w_p$ is the number of pixels in the mask. The final weight is calculated by:

$$w = w_d \times w_s \times w_p$$

# E  HYPERPARAMETER SENSITIVITY

**Context-aware embedding weights.**  For each 2D object mask we construct five CLIP image embeddings:

$$E_s, \ E_l, \ E_h, \ E_{\text{hide}}, \ E_{\text{mask}} \in \mathbb{R}^d,$$

corresponding respectively to (i) a small crop tightly around the object, (ii) a larger crop including more local context, (iii) a "huge" crop covering the object and its wider surround, (iv) a crop where the object is removed and only its surroundings are visible, and (v) an object-only crop obtained by masking out the background. Given these base embeddings, our normalized context-aware representation for an object is

$$E_{\text{ctx}} = E_s + \alpha_h \, E_h + \alpha_l \, E_l - \alpha_o \, E_{\text{hide}} + \alpha_m \, E_{\text{mask}},$$

where the scalar coefficients $\alpha_h, \alpha_l, \alpha_o, \alpha_m \geq 0$ control how much we emphasize wide-field context ($E_h$), medium-scale context ($E_l$), suppress misleading background-only evidence ($E_{\text{hide}}$), and sharpen the signal from the object-only view ($E_{\text{mask}}$).

**Sensitivity setup.**  In the following experiments we study the effect of these weights on 3D semantic segmentation performance. For each parameter $\alpha \in \{\alpha_h, \alpha_l, \alpha_o, \alpha_m\}$, we scale it by a factor $\lambda \in \{0.5, 0.75, 1.0, 1.25, 1.5\}$ while keeping the remaining coefficients fixed at their default values, and measure mIoU, mAcc, and fmIoU.

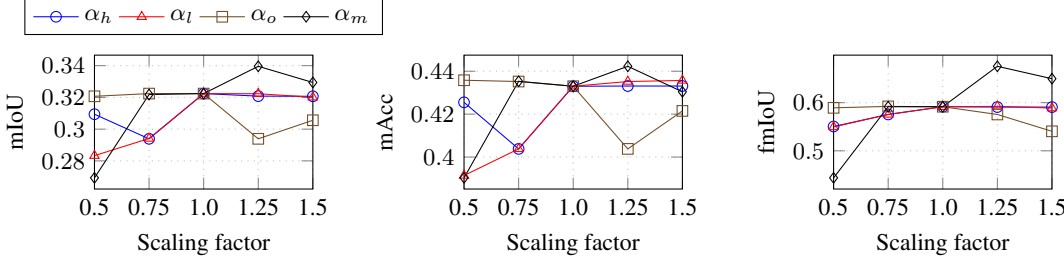

Figure 5: Hyperparameter sensitivity on Replica (scene `room0`) for the four weighting parameters $\alpha_h, \alpha_l, \alpha_o, \alpha_m$. Performance is reported in mIoU, mAcc, and fmIoU as the scaling factor is varied.

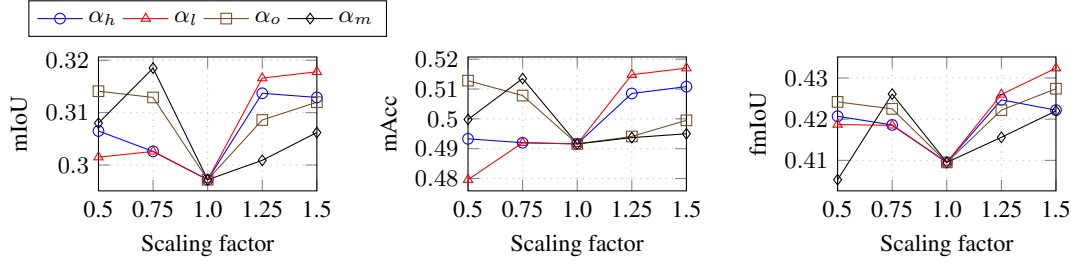

Figure 6: Hyperparameter sensitivity on ScanNet (scene `scene0011_00`) for the four weighting parameters $\alpha_h, \alpha_l, \alpha_o, \alpha_m$. Performance is reported in mIoU, mAcc, and fmIoU as the scaling factor is varied.

## F  VISUALIZATION

We include the following visualizations to enable a more detailed inspection of the results across different Replica scenes, facilitate comparison across methods, and to illustrate typical failure cases observed in challenging scenarios.

Ground Truth                                      Prediction

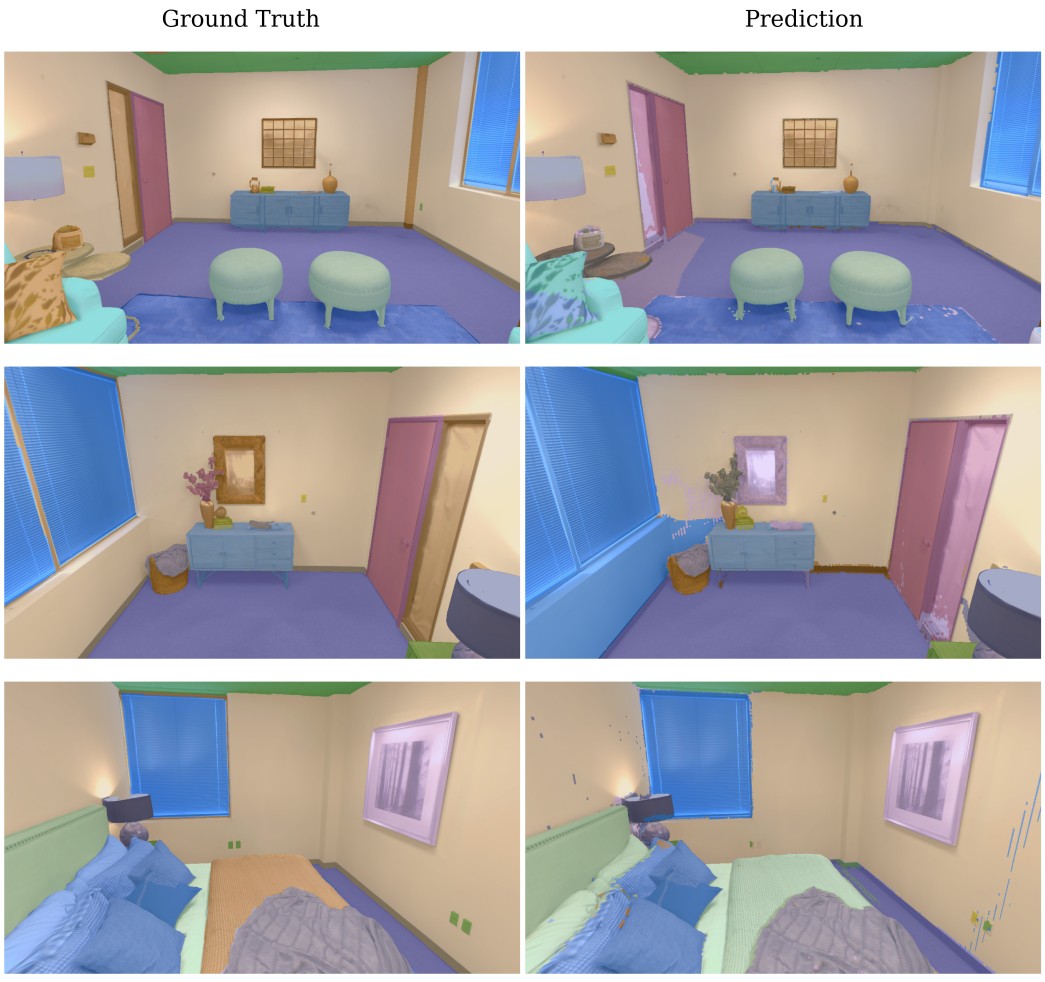

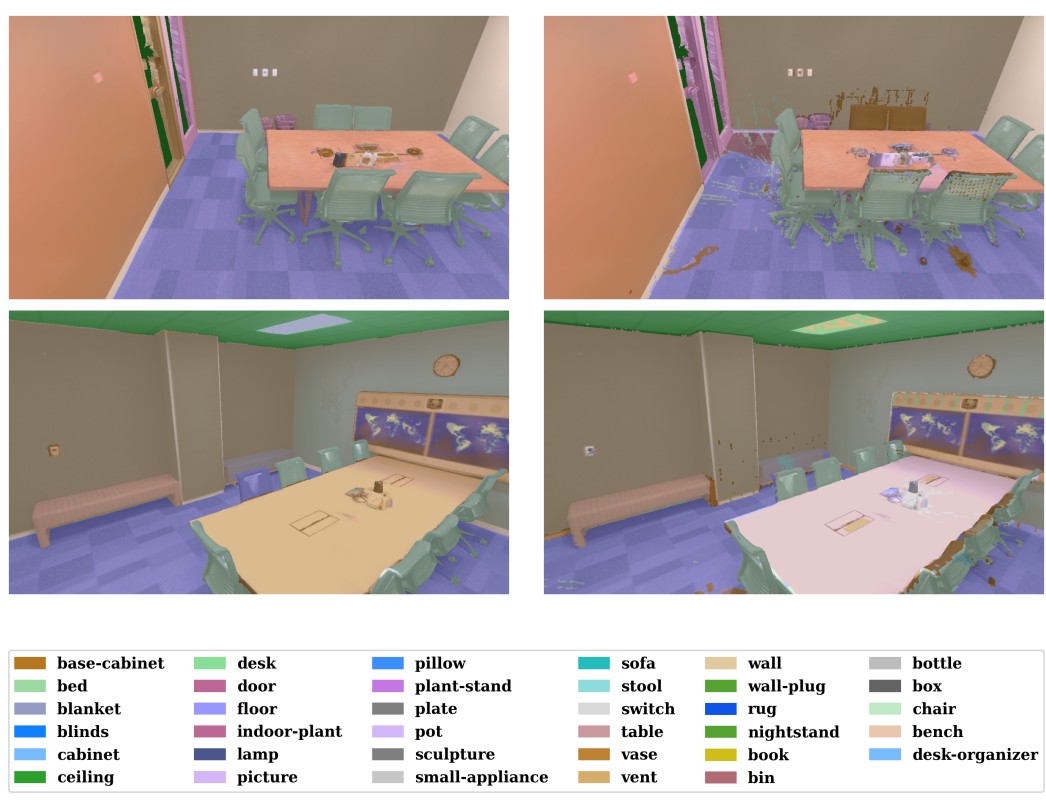

| | base-cabinet | | desk | | pillow | | sofa | | wall | | bottle |
|---|---|---|---|---|---|---|---|---|---|---|---|
| | bed | | door | | plant-stand | | stool | | wall-plug | | box |
| | blanket | | floor | | plate | | switch | | rug | | chair |
| | blinds | | indoor-plant | | pot | | table | | nightstand | | bench |
| | cabinet | | lamp | | sculpture | | vase | | book | | desk-organizer |
| | ceiling | | picture | | small-appliance | | vent | | bin | | |

