# OpenReview forum: "Context-Enriched Embeddings for Robust 3D Scene Understanding"
_ICLR.cc/2026/Conference — Submitted to ICLR 2026_

### Official Review · Reviewer_RZwy · 2025-10-24

**Soundness:** 2
**Presentation:** 2
**Contribution:** 1
**Rating:** 2
**Confidence:** 5

**Summary:**

Summary of the Paper
The paper introduces CORE-3D, a training-free pipeline designed for open-vocabulary 3D scene understanding and object retrieval based on natural language queries. The authors identify two key weaknesses in prior methods: 1) the generation of fragmented and incomplete object masks by standard segmentation models, and 2) the lack of sufficient visual context when assigning semantic labels, leading to inaccurate classifications.
To address this, CORE-3D proposes a multi-stage approach. First, it uses SemanticSAM with a progressive granularity strategy to generate high-quality 2D object masks. Second, it develops a context-aware encoding strategy that computes CLIP embeddings from five different contextual views of each mask and aggregates them. Finally, these 2D predictions are lifted to 3D, where they are merged and refined using geometric heuristics to ensure multi-view consistency. This results in a coherent 3D semantic map that can be queried using natural language to perform complex segmentation and retrieval tasks without any task-specific training.

**Strengths:**

Impressive Empirical Performance: The primary strength of this work lies in its impressive empirical results. The proposed method, CORE-3D, achieves state-of-the-art performance across multiple standard benchmarks, including Replica, ScanNet, and SR3D+. It significantly outperforms prior approaches in both 3D semantic segmentation metrics (mIoU, fmIoU) and language-based object retrieval accuracy. The strong quantitative and qualitative results provide compelling evidence of the method's effectiveness in complex 3D environments.
Well-Motivated Methodological Design: The paper effectively identifies and addresses key limitations of existing methods. The proposed context-aware CLIP embedding strategy, which aggregates features from multiple contextual crops, is a well-motivated and intuitive approach to mitigate semantic ambiguity caused by context-deficient object views. This design choice directly tackles a critical bottleneck in open-vocabulary 3D perception.

**Weaknesses:**

Lack of Ablation Studies and Unclear Contribution Attribution: The most significant weakness of this paper is the complete absence of ablation studies. The authors introduce several new components—a progressive SemanticSAM refinement strategy, a multi-crop context-aware embedding scheme, and 3D geometric refinement heuristics—but provide no experiments to disentangle their individual contributions. Consequently, the true source of the substantial performance gains remains unclear. This omission undermines the paper's methodological rigor and makes it difficult to verify the utility of each proposed component.
Limited Technical Novelty and Over-reliance on Existing Models: The technical novelty of the work is limited, as the framework is primarily a sophisticated integration of powerful, pre-existing models (SemanticSAM, CLIP, LLMs/VLMs). The performance heavily relies on the superior capabilities of SemanticSAM, which is not a contribution of this paper. The authors list a "SemanticSAM refinement strategy" as a main contribution, yet this appears to be more of a specific application of the model's inherent multi-granularity features rather than a novel algorithmic extension. Without a crucial ablation that compares its performance when using a more standard backbone (e.g., the original SAM), it is difficult to assess the true impact of the authors' own contributions versus the significant gains inherited from a superior off-the-shelf component.

**Questions:**

1.Regarding the lack of ablation studies: The performance gains demonstrated are impressive, but the lack of ablation studies makes it difficult to attribute these gains to specific components of your method. Could you provide experiments to quantify the contribution of each key component: (a) the progressive granularity refinement, (b) the context-aware CLIP embedding, and (c) the 3D mask merging/refinement? In particular, a crucial baseline would be to replace SemanticSAM with the original SAM within your pipeline to demonstrate the effectiveness of your proposed strategies independent of the advanced segmentation backbone.
2.On Hyperparameter Sensitivity: The paper states that the weights for the context-aware embedding and the overlap thresholds (τ_k) for mask generation are "empirically tuned." Could you provide more details on the tuning process? More importantly, how sensitive is the model's performance to these hyperparameters? A sensitivity analysis would significantly strengthen the claims of the method's robustness and aid in its reproducibility.
Analysis of Limitations and Failure Cases: The qualitative results are strong, but the paper would benefit from a discussion of the method's limitations. Could you provide some examples of failure cases? For instance, in what types of scenes or for what kinds of objects does the segmentation or retrieval pipeline struggle? An analysis of these failures could provide valuable insights for future work

---

> ### Author Response · Authors · 2025-12-03
>
> Dear Reviewer RZwy,
>
> Thank you for your thoughtful and constructive feedback. We appreciate your positive assessment of our empirical performance and have substantially strengthened the revised manuscript in response to your concerns.
>
> First, we have added a comprehensive set of ablation experiments (Section 5) that isolate the contribution of each component in our pipeline, including the mask-refinement module, the context-aware embedding mechanism, and the extension strategy. These results clearly demonstrate how each part contributes to the overall performance. To clarify the role of SemanticSAM, we additionally report comparisons among SAM, individual SemanticSAM granularity levels, and our progressive multi-granularity refinement. The experiments show that a single SemanticSAM level is unstable, and that most of the performance gains arise from our progressive accumulation and overlap-filtering strategy rather than from SemanticSAM alone.
>
> We have also incorporated a hyperparameter sensitivity analysis (Appendix E), which shows that the method remains robust across a wide range of values, indicating low sensitivity. Finally, in response to your request for limitations and failure cases, we provided qualitative examples in Appendix F. We believe these additions address your concerns and further clarify the contribution and robustness of our approach.

---

### Official Review · Reviewer_Bam8 · 2025-10-30

**Soundness:** 2
**Presentation:** 1
**Contribution:** 2
**Rating:** 2
**Confidence:** 3

**Summary:**

This paper introduces a method for building feature embedded 3D space. AUthors observe that prior works relies on inaccurate raw masks from SAM model, and object-only CLIP extraction that ignores surrounding context. To overcome, authors suggest multi-view consistent mask uplifting pipeline and context-aware CLIP features aggregation. For context aware CLIP feature, authors suggest to use multi-resolution image cropping from object to surroundings and weighted aggregation for context understanding. The paper also define a open-vocab object retrieval task and present a possible solution to handle such problem. Across experiments, the method shows improved context understanding compared to prior works.

**Strengths:**

- Proposes a new task (object retrieval) and presents one possible solution for the problem.

- Introduces a multi-resolution, context-aware strategy to strengthen context understanding—an aspect underemphasized in prior work.

**Weaknesses:**

**Heavy reliance on manual hyperparameters**
- Many steps require hand-tuned choices (cropping patches for context-aware CLIP, weighting for aggregation, filtering thresholds). These appear distribution- and scene-dependent, risking poor robustness when scene composition shifts.

**Missing ablations for design choices.**
- It is difficult to verify the benefit of each component:
- No direct evidence that multi-granularity crops improve context understanding.
- Other design choices are not empirically validated, making the claimed advantages hard to trust.

**Presentation & reproducibility (minor but notable).**
- Figures have tiny text and excessive margins; readability can be improved.
- The heading starting at line 319 should be bold.
- Reproducibility details are sparse (e.g., which retrieval model and which prompts for object retrieval). Including these (even in an appendix) would strengthen credibility.

**Questions:**

All questions are embedded within the Weaknesses section above.

---

> ### Author Response · Authors · 2025-12-03
>
> Dear Reviewer Bam8,
>
> Thank you for your assessment and constructive reviews. We have updated the paper to address all concerns.
>
> Our hyperparameters are chosen based on evaluations on the validation set of the ScanNet dataset. These parameters lead to strong performance on both the ScanNet and Replica test subsets. We also include a sensitivity analysis (Appendix E) where each of the CLIP aggregation weights is varied over a range of scaling factors. The results show that performance is stable around the chosen values.
>
> Regarding missing ablations, We have added an ablation study section to the paper. In this section, you can examine how the presence or absence of each component in our pipeline affects the results, including the effectiveness of the multi-granularity mask generation. We also compare the context-aware CLIP component with OvSeg, which is a fine-tuned version of CLIP for mask inputs. Furthermore, we compare our cropping method for generating different context views of masks with an alternative method that does not require fixed hyperparameters for extension, and we show that our approach achieves better results.
>
> Details of the object-retrieval framework, including the utilized models and prompts, are provided in the appendix. We also improved figure readability and fixed minor formatting issues.

---

### Official Review · Reviewer_ENYB · 2025-10-31

**Soundness:** 3
**Presentation:** 3
**Contribution:** 1
**Rating:** 2
**Confidence:** 4

**Summary:**

This paper proposes an algorithm for building a semantic point embeddings. Given posed images and points, this paper first extract the class-agnostic masks from images. Second, for each view, this method compute CLIP embeddings while considering the different size of bounding boxes that potentially have more context information. Last, the method merges 2D masks into 3D masks and locate the CLIP embeddings on the 3D masks.

**Strengths:**

The overall flow of this paper is readable and understandable. The proposed schemes are reasonable to build up the semantics embeddings on top of pointclouds.

The proposed pipeline is quite similar to the ConceptFusion, but this paper addresses the problem of SAM and resolve this by SemanticSAM, which makes sense to me. Moreover, different size of crops for extracting CLIP embeddings looks good to me as well. While it can be a naive solution, the addressed problem stated in Line 242 of the manuscript is true.

While the paper achieves performance improvement compared to previous studies, I found many issues.

**Weaknesses:**

__W1. Weak comparison__
I believe that the authors mostly track the related studies that are originated from ConceptFusion, ConceptGraph, etc. However, in the recent studies, such as Mosaic3D [A] and RegionPLC [B], these studies also tackle the same task, open vocabulary 3D semantic segmentations. Moreover, if the authors refer to the line of this studies, these methods also utilize the Vision Language Models for their data generation pipeline where the generated data are used to train the 3D neural networks.

Despite the high similarities, the authors did not cite the papers. Accordingly, there is no technical comparisons with this submission and [A,B]. Thus, I cannot be sure whether this paper really achieves high-quality predictions in comparison with [A,B] as well. The authors should have checked the related studies a lot.

__W2. Mask merging is not something new__
I recommend the authors to refer to this paper, Gaussian grouping [C]. This paper aims to obtain the view-consistent masks (which is called `mask merging` in this submission) by leveraging 3D Gaussian Splatting [D]. In my understanding, this submission as well as [C] starts from the similar input data: images, camera parameters (extrinsic / intrinsic). Within the detail, [C] also computes the overlapping ratio between masks and 3D Gaussians as proposed by this submission.

Based on this understanding, I do not think that the proposed mask merging described in Section 3.3 is novel or unique. Moreover, there is no qualitative / quantitative analysis on this module, so I cannot catch its effectiveness as well.

__W3. What is the template to use VLM?__
In Section 3.4, the authors use VLM to perform the object retrieval task. Commonly, when the papers use the VLM or LLM, the papers provide the templates to be used for their target task. However, in this submission, I cannot find such a information throughout the manuscript. This is quite a serious problem.

__Related works__
[A] Mosaic3D, CVPR 2025
[B] RegionPLC, CVPR 2024
[C] Gaussian grouping, ECCV 2024
[D] 3D Gaussian Splatting for Real-Time Radiance Field Rendering, SIGGRAPH 2023

**Questions:**

Overall, the paper is not ready for the submission. I recommend the authors to check the relevant and the recent studies about pointcloud based OpenVocab segmentation methods. Moreover, if the authors look at the 3D Gaussian based methods, such as LangSplat[H] and OpenGaussian [F], the authors can find the numerous ways of extracting CLIP embeddings from images.

Nonetheless, I hope the authors' responses for my opinions and analysis.

__Related works__
[F] OpenGaussian, Neurips 2024
[H] LangSplat, CVPR 2024

---

> ### Comment · Reviewer_ENYB · 2025-11-26
> **No rebuttals?**
>
> The deadline for the rebuttal period is coming. I hope to see the rebuttals. If the authors are still working on it, please let me know.

---

> > ### Author Response · Authors · 2025-11-26
> >
> > Dear Reviewer ENYB,
> > We are working to finalize the revised version of our paper to resolve the mentioned issues by the reviewers. Specially, we are working on the ablation study part to clarify the effect of each component in our pipeline. We estimate to submit the new version untill Nov 28.

---

> ### Author Response · Authors · 2025-12-03
>
> Dear Reviewer ENYB,
>
> Thanks for your assessments and analysis. We have updated the paper to address all concerns.
>
> The main difference between our method and the papers Mosaic3D [A] and RegionPLC [B] is the input data. Our pipeline processes RGB-D images along with the camera poses from which those images were captured, whereas those methods have access to 3D ground-truth point clouds. Therefore, they train their models to assign labels directly to those points, but in our framework we must rely on RGB-D images and camera poses. For example, for the Replica dataset, we use the RGB-D and camera-pose data released by NiceSLAM, which performs a SLAM model on the 3D mesh dataset and has also been used by prior works. Of course, the output of SLAM models is not perfect, and their inaccuracies accumulate as error in our performance.
>
> Also, our model can be used for real-world scenarios where no ground-truth 3D point clouds are available, whereas such models cannot. Thus, we believe that it is not fair to compare our method with Mosaic3D [A] and RegionPLC [B]. However, for OpenGaussian [H] and LangSplat [L], in the edited version of our paper for rebuttal, we have added these papers’ results in the experimental section to demonstrate our model’s performance compared to theirs. We have also added the results of LERF and OpenNeRF (accepted to ICLR 2024).
>
> For the mask-merging section, we think that the averaging formula in our paper was misleading. We intended to elucidate the entire mask-merging framework in the appendix, but due to lack of time we did not submit the appendix during the initial submission. We have now added the appendix section in the edited version, so the reviewers can refer to it to assess the whole mask-merging framework.
>
> We have also added several ablation studies to our paper and demonstrated the effectiveness of each component in our pipeline.
>
> Thanks for your consideration.

---

### Meta-Review · Area_Chair_VdHE · 2025-12-17

**Summary:**

This paper proposes an algorithm for building a semantic point embedding. Given posed images and points, this paper first extracts the class-agnostic masks from images. Second, for each view, this method computes CLIP embeddings while considering the different sizes of bounding boxes that potentially have more context information. Last, the method merges 2D masks into 3D masks and locates the CLIP embeddings on the 3D masks.

All reviewers are negative about this submission, while the authors fail to provide a sufficient point-to-point response. Therefore, the final decision is Reject.

**Reviewer Concerns:**

Key concerns:
1. Lack of novelty.
2. Experimental comparison is insufficient.
3. Missing ablation study.
4. Hyperparameters are labor-intensive.

**Reviewer Scores:**

The reviewers will keep their scores.

---

### Decision · Program_Chairs · 2026-01-26

Reject